# Land suitability analysis in monocentric post-socialist city: Case of Ulaanbaatar, Mongolia

**Galmandakh Boldbaatar**[1‡], **Gantulga Gombodorj**[1‡], **Dorligjav Donorov**[1¤a], **Robert Andriambololonaharisoamalala**[2], **Myagmarjav Indra**[3¤b]*, **Myagmartseren Purevtseren**[1]*

1 Department of Geography, Research Laboratory of Land Planning and Survey, National University of Mongolia, Ulaanbaatar, Capital City, Mongolia, 2 School of Earth and Planetary Sciences, Curtin University, Perth, Western Australia, Australia, 3 Department of Land Management, Mongolian University of Life Sciences, Ulaanbaatar, Capital City, Mongolia

¤a Current address: Department of Geography, School of Arts and Scineces, National University of Mongolia, Ulaanbaatar, Capital City, Mongolia
¤b Current address: Department of Land Management, School of Agro-Ecology, Mongolian University of Life Sciences, Ulaanbaatar, Capital City, Mongolia
‡ GB and GG are contributed equally to this work and Joint Senior Authors
* myagmarjav@muls.edu.mn (MI); land_management@num.edu.mn (MP)

**Data Availability Statement:** All relevant data are within the manuscript and its Supporting Information files.

## Abstract

Urban expansion has been rapidly increasing and is projected to be tripled in 2030 in world-wide. The impact of urbanization has adverse effects on the environment and economic development. Residential lands consist of almost one-third of the urban area and heavily affect the city's inhabitants. The capital of Mongolia, Ulaanbaatar, has been significantly expanded, particularly in the urban periphery where poor living conditions and a lack of essential urban services were identified. The paper aims to conduct a suitability analysis of residential areas in Ulaanbaatar city based on three main categories (livability, affordability, and accessibility) of fifteen criteria using the fuzzy logic. Through the study, we have identified some potential suitable residential areas for further development, such as apartment residential area located in the southern part of the city and four low-rise ger areas were distributed along major transport corridors. Moreover, the results indicated that the spatial structure of the whole town might be in transition to a polycentric pattern. However, a concentric ring pattern in the 'city's periphery displayed a concerning uncontrolled ger area expansion, which may increase low living conditions in the area. This study recommends better urban sprawl control policies and more property market investment in the ger area to ensure sustainable development goals in Ulaanbaatar.

## 1. Introduction

According to the UN, the pace of urbanization is projected to be the fastest in many developing countries, facing challenges of uncontrolled urban growth and unstructured urban sprawl [1]. One prime example is capital city of Mongolia. The city's population has increased fifteen times in the last few decades due to rapid rural-urban migration after the democratic

**Funding:** This research was funded by the National University of Mongolia (NUM), research project and grant number: P2022-4385 and P2023-4621. The funders had no role in study design, data collection and analysis, decision to publish, or preparation of the manuscript.

**Competing interests:** The authors have declared that no competing interests exist.

revolution [2]. It has totally changed the pattern of the city to ger area expanding as 51.8% [3]. The impact of urbanization is adverse, creating environmental, social and economic problems globally [1,4,5]. Purevtseren et al. noted that air pollution, soil pollution, traffic condition, lack of essential services and poverty were serious problems in Ulaanbaatar city [6]. Moreover, severe damage to water resources was predicted due to city expansion [7]. The Ulaanbaatar city is the heart of the country, contributing 65% of the 'country's GDP, 85% of electricity power, and 50% of investments [2]. The uncontrolled urban sprawl in the Ulaanbaatar city may slow down the economic development and increase the environmental "footprints" of the country.

In 1933, 'Athens Charter' put forward a vision that residential land development is the top priority of a city, which is the heart of a functional town [8]. The uncontrolled residential land development drove many urban sprawl issues. For instance, Purevtseren et al. [3] found that 32% of the area in Ulaanbaatar city was covered by residential "ger" area (sprawl). According to the Japan International Cooperation Agency (JICA)'s questionnaire survey report, residents of Ulaanbaatar city have risen that security and environment were the most critical concerns of their sustainable life [7]. The essential aspect of a suitable and sustainable living environment in urban areas is good air quality and health and welfare services. Moreover, citizens commonly requested to locate the healthcare facilities near residential areas [7].

Furthermore, the typical monocentric spatial arrangement of the city was increasingly equipped in the city center throughout the Soviet era [9]. A significant population increase following the democratic revolution in 1990 created unregulated pressure on the city's centralized spatial structure. Additionally, Bertaud [10] examined the spatial organization of cities in Central and Eastern Europe (CEE) and discovered that these regions have a monocentric spatial structure. Following market transitions, it observed a gradual transition to polycentric structures. Despite limited research on the monocentric spatial structure of Ulaanbaatar city, there is a noticeable presence of this kind of transition in this area.

Urban planning is consistently linked to a variety of decision-making difficulties and strategies, and the results of planning will differ based on the combination of the approaches and techniques employed. During the Soviet era, Mongolia, like many other communist nations, employed planning approaches and techniques to design cities without involving citizens. This resulted in a simple, quick, and interruption-free planning procedure during the project [11].

In our knowledge, in the Soviet era, a monocentric structure allowed the communist government to "control everything" related to urban management. According to the centralized economy, it was cost-effective and easy to manage particularly to build mega infrastructure and new residential district. The urban development process was implemented under "bulldozer urbanism" approaches, whereas all land had been owned by the state. There was no need for residents' consent or negotiations with them to expropriate their lands. The municipality directly canceled the resident permit and gave notice to vacate the new development area by official declaration. This procedure referred to as "enter to ground" characterizes communist regimes' relation to their people. For the new development areas, only the land capability for construction was considered, focusing on investigating the physical characteristics of ground soil and topography.

These days, many of these nations have new laws requiring public engagement while also using new approaches and technologies. One of the most difficult tasks facing communities and municipal governments today is to locate prospective locations for the sustainable growth of city subcenters without threatening human rights and environmental protection. Land suitability analysis offers tools to address these management problems [11].

Ulaanbaatar city's monocentric structure hinders sustainable development promoting traffic jams, air pollution, land grabbing, etc. Several international and local experts suggested to

replan and switch from monocentric to polycentric structures by developing new city subcenters in Ulaanbaatar [7,11,12].

In short, there has been a constant increase in interest in adopting land suitability analysis since it can handle and combine many types of data (both spatial and non-spatial, as well as multi-temporal and multi-scale) in an efficient manner. Then, from a technical perspective, comprehension of the fundamental issue, and democratic points of view can all benefit the planning process [11].

Therefore, it is now widely admitted that analyzing and identifying the suitability of residential areas is very important for appropriate land use policies to improve the living condition of citizens for future urban development [13–15].

To support sustainable urban development, a scientific and systematic approach for evaluating and planning the residential area development is essential. However, very limited research work in Ulaanbaatar city has been done. Such research may help the public, such as citizens justify their residential selection and real estate agencies, and land developers and investors make appropriate residential land development decisions.

The process of land suitability analysis involved multicriteria analysis (MCA) and geographic information system (GIS), and the land suitability evaluation methods have significantly improved due to the development of data science and GIS. Multicriteria analysis manages complex factors for suitability analyses and has been used in various industries [16–19]. The integration of GIS and MCA takes advantages of the professional field with publicity, productivity, efficiency, and accuracy [20].

However, determining the appropriate criteria is an essential and complicated step to achieving the scientific evaluated results [21,22]. The criteria of residential suitability analysis are multidimensional, generally including socio-economic and environmental factors, and their subdivisions into more exact criteria depending on the specification of research [23]. Commonly, natural conditions including slope, aspect, elevation, lithology, waterbody, green space and earthquake; and socio-economic conditions, such as demography, community accessibility, employment, housing, health, and safety, are mainly employed for residential suitability analysis [13,17,23]. However, the data availability and quality were mentioned as long-lasting challenges. As for popularity in this research field, a number of scholars demonstrated the integration of evaluation criteria as effective to develop the factors in the case study [22].

Many case studies have used MCA methods that include the ranking of the degree of strength and weakness by using the analytic hierarchy process (AHP) [11], ordered weighted averaging (OWA) [17,24,25], Bayesian networks [26], matter element [27], artificial neural network [28], agent-based model [14] and fuzzy logic theory [13,21,22,29,30]. Even though the usage of the MCA method is more popular, it still has several concerns. Firstly, data preparation and data quality are the issues that can heavily affect the results. The standardization of criteria is the next barrier that has more concern on mathematical simulation. Moreover, problems dependency of experts is remaining, such as the subjective uncertainty and some mathematical calculation for the standard quantification of evaluation factors [31–33]. Furthermore, in working with multiple criteria, it is complicated since the importance of criteria may vary. To minimize the subjectivity and mathematical shortcomings, the fuzzy logic theory, which was initially developed by Zadeh [34], was highly valued in the latest studies [24,35,36]. However, determining the membership of criteria, selecting fuzzy membership, and the criteria overlaying method still involve a certain level of subjectivity. Qualified references and validation were recommended to control the errors. The fuzzy logic theory is common in land suitability analysis, particularly residential suitability [29]. However, this method has not been used in the residential suitability and urban development industry in Ulaanbaatar.

Thus, for this study, we aim to develop a new framework for analyzing the residential suitability of Ulaanbaatar city using spatial analysis techniques (fuzzy logic theory), to evaluate the current residential suitability status and to provide commendation for future sustainable residential development. Moreover, the developed methods and outputs of this study may facilitate Ulaanbaatar urban development agencies to develop sustainable residential development policies and help citizens make a right residential location choice. The following objectives will be fulfilled in order to do this:

i. Identify factors in case of Ulaanbaatar city based on literature review;

ii. Conduct suitability analysis using fuzzy AHP and Multi criteria analysis methods;

iii. Validate the developed model and recommend suitable areas for further residential development regarding Sustainable Development Goals (SDGs).

## 2. Materials and methods

### 2.1. Study area

Ulaanbaatar city is the Mongolian capital, located in the north-central part of Mongolia (Fig 1). It is the country's cultural, industrial, and financial hub, and the centre of Mongolia's road network [3]. Ulaanbaatar covers approximately 4,740.4 km2, the equivalent of 0.3% of the total area of Mongolia (1.57 million km2). Ulaanbaatar has six central and three remote districts, and it is divided into 171 suburbs. It is the most populated city of Mongolia with approximately 1.8 million inhabitants, which accounts for 54.5% of the total population of Mongolia, with a population density of 308 people per km$^2$ [3,11]. Ulaanbaatar is the world's coldest

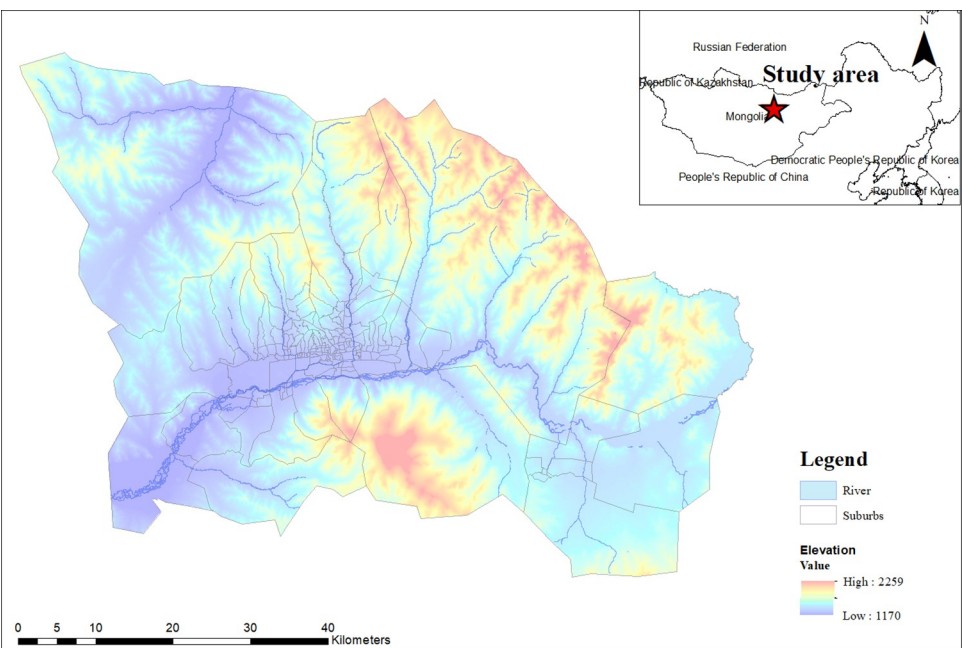

**Fig 1. Location of the study area.** Data sources: DEM (elevation) based on Advanced Spaceborne Thermal Emission and Reflection Radiometer (ASTER) [39] (Available online: http://www.jspacesystems.or.jp/ersdac/GDEM/E/), river, and suburbs produced by the General Authority for Land administration, Geodesy and Cartography of Mongolia (ALMGC) [40] (Available online: https://nsdi.gov.mn/open-layer/). All sources are in the public domain and not copyrighted.

capital city, with an annual mean temperature of −1.3˚C with its harsh winter climate minimum temperatures frequently falling below −30˚C [37,38]. Geographically, the city is at an elevation of about 1,350 meters in a valley on the Tuul river, surrounded by mountains. Even though Ulaanbaatar city is not the largest capital cities globally (Fig 1), it has been significantly expanded since the mid-twentieth century. For instance, in 1956, 118,000 residents were recorded in Ulaanbaatar [3].

## 2.2. Dataset

In this study, 25 types of data provided by The General Authority for Land administration, Geodesy and Cartography of Mongolia (ALMGC) [40], Research Laboratory of Land planning and survey, National University of Mongolia (NUM-Lab.RaS) [41] and National Statistics Office of Mongolia (NSO) [42] were used. Data of landscape for an illustration base map on all figures, particularly slope degree, was gathered from a subset of an optical stereo-based ASTER GDEM elevation data as well as radar-based SRTM elevation data from the CGIAR Consortium and the United States National Aeronautics and Space Administration (NASA) and the Ministry of Economy, Trade, and Industry (METI) of Japan [39,43]. Other sources for data and information inputs and figure representation belong to public organizations (ALMGC [40], NSO [42] and JICA [7]) or our laboratory (NUM-Lab.RaS) [41] listed below. The administration boundary, land use map, land price, road networks, heating supply network, water supply network, sewerage network, electricity network, railroad, land price and cadastral parcel and building were provided by ALMGC open-spatial data online portal [40], which were obtained through the authorized access rights by the authors. The data of schools, kindergartens, hospitals, bus stops, and air pollutions were collected from database of the NUM-Lab.RaS [41]. Finally, the census data were generated from the open data of NSO's website [42].

Based on the literature and data availability, 15 different criteria were developed and processed in ArcGIS from ESRI (http://www.arcgis.com) platform (Table 1).

**Table 1. Data layers used in suitability analysis.**

| Data layer | Source | Format |
| --- | --- | --- |
| **1. Slope (degree)** | SRTM & GDEM [39,43] | Raster |
| **2. Crime rate (%)** | NSO [42] | Raster from Vector & Census data |
| **3. Land use (code)** | ALMGC [40] | Raster from Vector |
| **4. Facility cover (%)** | ALMGC [40] | Raster from Vector |
| **5. Building cover (%)** | ALMGC [40] | Raster from Vector |
| **6. Land price (□ per square m)** | ALMGC [40] | Raster from Vector |
| **7. Distance from main park (m)** | ALMGC [40] | Raster from Vector |
| **8. Distance from air pollution point (m)** | NUM-Lab.RaS [41] | Raster from Vector |
| **9. Distance from water protection zones (m)** | ALMGC [40] | Raster from Vector |
| **10. Distance from school (m)** | NUM-Lab.RaS [41] | Raster from Vector |
| **11. Distance from kindergarten (m)** | NUM-Lab.RaS [41] | Raster from Vector |
| **12. Distance from hospital (m)** | NUM-Lab.RaS [41] | Raster from Vector |
| **13. Distance from bus stop (m)** | NUM-Lab.RaS [41] | Raster from Vector |
| **14. Distance from shopping centre (m)** | ALMGC [40] | Raster from Vector |
| **15. Distance from central and sub-central business district (m)** | JICA [7] | Raster from data (digitized) |

## 2.3. Methodology

**2.3.1. Defining residential suitability criteria.** The literature study of urban suitability analysis had carried on to select the necessary criteria for the study (Table 2). According to the literatures and data availability, we categorize the 15 criteria into three types: livability, affordability, and accessibility. Slope, main parks, air pollution, water protection zones, crime rate, and land use belong to livability criteria. School, kindergarten, hospital, bus stop, shopping centre, central and sub-central business district (CBD), facility cover, and building cover were categorized as accessibility criteria. Land price with regards to affordability, the was the indicator.

**Table 2. Criteria sources.**

| | Author | Criteria | Key findings |
|---|---|---|---|
| 1 | **Malczewski and Rinner, 2005 [17]** | 57 "objective" indicators categorized into eight domains: population resources, community affordability, community stress, community participation, employment, housing, health, and safety. | Ordered Weighted Averaging (OWA) procedures are most useful in situations involving a large number of evaluation criteria. Emphasized the importance of using the quantifier-guided OWA in conjunction with an exploratory multicriteria analysis |
| 2 | **Samarakoon et al., 2016 [25]** | Slope, landslide susceptibility, water reservation, water availability, mineral resources, road reservation, road accessibility, soil type, hazard prone areas, forest reservation, service centers, land use, archaeological sites, and community participation. | The integration of the assessment process and geographic information system technique gives a contribution to decision-making while minimizing social and environmental impacts. |
| 3 | **Qiu and Zhang, 2011 [44]** | Topography, geology, living environment, living facility & service, cultural & educational facility, municipal infrastructure, comprehensive management, planning & design, location, internal traffic | Proposed "Urban Residential Land Suitability (URLS) Evaluation index system" which consists of 32 indicators based on Fuzzy Comprehensive Evaluation and helps to assess existing residential area or found suitable sites for planning. |
| 4 | **Samad and Morshed, 2016 [45]** | Accessibility of service and facilities: near to existing settlement, school proximity, distance of park and playground, road proximity of paved and semi-paved, proximity of bus stops, distance to major highways. Socio-economic aspects: land value. Environmental aspects & safety: flood prone area, industry Topographical aspect: slope | GIS-based methods provide a more feasible system and attainable objectives on a biased-free basis for making decisions on site selection. |
| 5 | **Patil et al., 2012 [46]** | Water Availability, flood line distance, air pollution data, WQI, distance from waste disposal site | The Analytic Hierarchy Process method was found to be a useful method to determine the weights, compared with other methods used for determining weights |
| 6 | **Wang et al., 2021 [14]** | Natural factor—evaluation index, elevation, slope, geomorphic type, Socioeconomic factor—distance from highway, provincial road, national road, railway; land use type, Ecological protection factor—public administration, commercial, public service agency service point kernel density, distance from permanent basic farmland, distance from the lake, euclidean distance from the river, vegetation coverage, distance from nature reserve | By using logistic regression, it could accurately evaluate the effects of a single factor, thereby avoiding subjective assessments. |
| 7 | **Ustaoglua and Aydınoglub, 2020 [22]** | Aspect; elevation; distance from highways, bus stop, metro stop, port, airport, reservoirs, water courses, coastline, urban green, industry/commerce, residential centers; land use; soil capability | The findings of spatial analysis have considerable potential for urban development. |
| 8 | **Li et al., 2018 [31]** | Human activities: existing land use, pollution source, significant infrastructure, road traffic, Historical sites: ancient and famous tourism resources, heritage sites, Natural landforms: water area, slope, elevation, heat island effect, geological disasters Biological protection: ecological patches, NDVI | Urban Green Space suitability evaluation method is optimal, as it fully takes the advantage of the subjective weight method while avoiding the deficiencies of the objective weight method, thus keeping a good balance between subjectivity and objectivity |
| 9 | **Bathrellos et al., 2017 [47]** | Lithology, distance from active faults, slope, rainfall, land use, distance from roads, distance from streams | The proposed methodology gave reliable results, regarding the actual hazard events and the suitability for urban development maps |
| 10 | **Cardone and Di Martino, 2021 [13]** | Population, education, transportation, services | GIS-based hierarchical fuzzy takes advantage of complex data management evaluates land parcels and facilitates the attribution of the degrees with land parcels meet criteria. |
| 11 | **Liu et al., 2014 [24]** | Geology, geomorphology, hydrology, ecology, sociology, and economics | The methodology and the results of the study presented considerable recommendations to improving the long-term urban development plans of Beijing. |

**Table 3. Parameters used in the analysis of residential suitability.**

| Main criteria | Sub-criteria | | | | | |
|---|---|---|---|---|---|---|
| **Criteria classification** | | | | | | |
| | | 5 | 4 | 3 | 2 | 1 |
| | | High | Moderate | Medium | Low | Very Low |
| *Livability* | 1. Crime rate (%) | *0–0.6* | *0.7–1.2* | *1.3–2.0* | *2.1–3.2* | *3.3–4.4* |
| | 2. Land use (code) | *11* | *61* | *24 & 23* | *21–23* | *13–16, 26–42, 63–65* |
| *Accessibility* | 3. Facility cover (%) | *0.18–4.18* | *4.19–9.42* | *9.43–15.60* | *15.61–23.07* | *23.08–38.87* |
| | 4. Building cover (%) | *0–8.2* | *8.2–21.2* | *21.2–43.4* | *43.4–96.4* | *96.4–176.9* |
| *Affordability* | 5. Land price (□ per square m) | *< 48808* | *48809–95549* | *95559–180393* | *180394–505308* | *505309–857143* |
| **Fuzzy membership functions** | | | | | | |
| | | Upper limit | Lower limit | Fuzzy functions | | |
| | 6. Slope (degree) | *20* | *0* | *Decreasing linear* | | |
| *Livability* | 7. Distance from main park (m) | *10000* | *0* | *Decreasing linear* | | |
| | 8. Distance from air pollution monitoring point (m) | *0* | *5000* | *Increasing linear* | | |
| | 9. Distance from water protection zones (m) | *50* | *0* | *Boolean* | | |
| *Accessibility* | 10. Distance from school (m) | *3000* | *0* | *Decreasing linear* | | |
| | 11. Distance from kindergarten (m) | *3000* | *0* | *Decreasing linear* | | |
| | 12. Distance from hospital (m) | *5000* | *0* | *Decreasing linear* | | |
| | 13. Distance from bus stop (m) | *1000* | *0* | *Decreasing linear* | | |
| | 14. Distance from shopping center (m) | *2000* | *0* | *Decreasing linear* | | |
| | 15. Distance from central and sub-central business district (m) | *5000* | *0* | *Decreasing linear* | | |

Data sources: Crime rate were generated from the open data of NSO's website [42], land use, facility cover, building cover, land price, distance from main park, distance from water protection zones, distance from shopping centre were provided by ALMGC through the open-spatial data online portal [40], distance from air pollution point, distance from school, kindergarten, hospital, bus stop are processed data of the NUM-Lab.RaS [41], slope degree information was gathered from a subset of an optical stereo-based ASTER GDEM elevation data as well as radar-based SRTM elevation data [39,43], distance from CBD information based on JICA study [7].

Regarding livability, the slope is the factor that can increase construction costs and the risks of soil erosion and landslide, and a slope smaller than 20 degrees was considered to be potential of any usage [20]. In Ulaanbaatar city, green space is scarce, within 10 kilometers and closer to the main park were considered to be an appropriate distance [2]. Air pollution of the study area represents PM10, and PM2.5 as the pollution levels from air pollution source points, and literature suggested that air pollution in Ulaanbaatar city was eight times higher than the safest level of World Health Organization (WHO). It revealed that the spatial pattern where remote area from high level were less polluted [38]. Environmentally, water is a serious concern, and within 50 m from water bodies, no construction activities are allowed, and it is also not suitable for residential and commercial usage [22]. The crime maps data are categorized using a modified version of the Jenks natural breaks classification method, aiming to minimize within-group variance and maximize between-group variance [48]. Crime rate shows a percentage of recorded crime per population in each suburb and was categorized into five classes from low to high. As for the land use (ALMGC's land use map), undeveloped (land use code ""21"") and developed residential area (land use code ""22""), public land (land use code ""24""), recreational land (land use code ""25""), protected area (land use code ""61"") and open space (land use code ""11"") were selected and categorized into five classes (Table 3).

In terms of accessibility, school, kindergarten, hospitals, bus stops, shopping centre, central and sub-central business districts (CBD) have the highest score since they are essential for the residents [2,22]. Infrastructure and building density were categorized into five classes. Facility cover represents road network, heating supply network, water supply network, sewerage

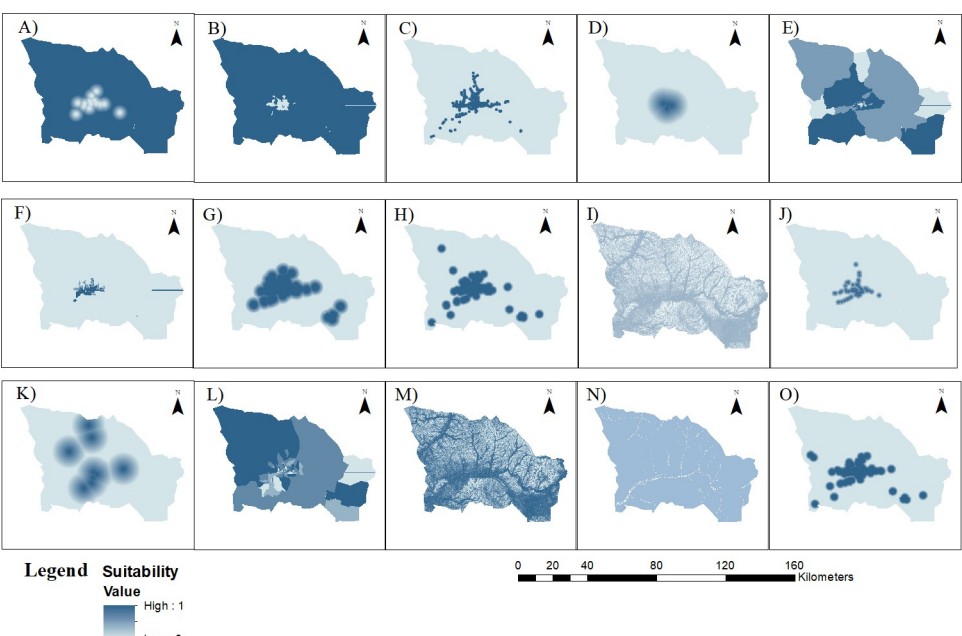

**Fig 2. Criteria maps for the residential analysis.** Noted: (A) Distance from schools, (B) Water restriction zone, (C) Slope, (D) Land price, (E) Distance from the main parks, (F) Distance from the shopping, (G) Land-use type, (H) Distance from kindergartens, (I) Distance from hospitals, (J) Facility cover, (K) Crime rate, (L) Distance from CBD, (M) Distance from bus stops, (N) Building cover, (O) Distance from air pollution points. The criteria maps were created by the authors in QGIS (https://qgis.org/en/site/). Data sources: criteria map A, H, I, M, O data of the NUM-Lab.RaS [41]; B, D, E, F, G, J, N were provided by ALMGC [40] (Available online: https://nsdi.gov.mn/open-layer/); C from a subset of radar-based SRTM data [43] (available online https://srtm.csi.cgiar.org); K open data of NSO [42] (Available online: https://data.nso.mn/datamart); L based on JICA study [7].

network and railroad network, and it reveals the percentage of cover on each suburb, and high level of coverage indicates high suitability [7]. Regarding building coverage, it was calculated the same as facility cover; however, a high level of building cover indicates a low score of suitability because the higher building density, the less potential for further development. A limitation for future development, such as a new house, infrastructure, green area etc.

Regarding land price, it was also classified into five categories, and lower price is suitable due to elevated cost for development [49].

**2.3.2. Commensuration of the criteria.** To conduct residential suitability analyses, measurement scales of the criteria were required to be a competitive unit. We normalized criteria value ranging from 0 to 1. A number of different approaches were used in the literature, including deterministic, linear standardization and fuzzy logic theory. In this case study, we used the deterministic and fuzzy logic methods. In terms of the conventional deterministic approach, we rescaled criteria using a score range from 0 to 255. The crime rate, land use, facility cover, building cover and land price were classified into five categories representing suitability from very low (1) to high (5). Each class was rescaled as follow: ""Very low"" as 50, ""Low"" as 100, ""Medium"" as 150, ""Moderate"" as 200 and ""High"" as 255, and it was normalized by the maximum value (Table 3). All rescaled criteria were simply normalized to be a competitive unit from 0 to 1 (Fig 2).

Secondly, in classical set theory, membership and boundary are strict, defined as true or false that are the members of a set [50]. Regarding fuzzy set theory, fuzzy boundary was defined, which has a gradual transition between two zones [51]. Fuzzy membership is a continuous grade from 0 (non-membership) to 1 (full membership). It is mathematically formulated

as follows [50]:

$$A = \{(y, \mu_A(y)) | y \in U\} \tag{1}$$

$$\mu_A(y) \in [0, 1] \tag{2}$$

where A is a fuzzy set, and it is given as where y belongs to the area U and μ_A (y) is a degree of membership y in a fuzzy set A while y is an element of fuzzy set A. μ_A (y) Membership value is a degree between 0 to 1. Fuzzy sets are represented as membership functions. In the literature, a variety of functions are used, including sigmoidal, J-shaped, linear, and user-defined [50]. The value 1 of the membership indicates the maximum suitability, whereas a score of 0 is the lowest suitability for residential usage. This study used fuzzy linear membership function to transform the input data 0 to 1 scale. It is formulated as follows [50]:

$$\mu_A(y) = \begin{cases} \dfrac{y_2 - y}{y_2 - y_1} & for\ y_1 < y < y_2 \\ 0 & for\ otherwise \end{cases} \tag{3}$$

All the classification methods of criteria are based on literature sources in Table 2 and below presented criteria classifications in detail (Table 3).

**2.3.3. Overlaying of the criteria.**   Zimmermann and Zysno [52] discussed a variety of fuzzy overlay operations and highlighted five useful operators to combine independent datasets with the fuzzy AND, fuzzy OR, fuzzy algebraic product, fuzzy algebraic sum and fuzzy γ -operators. Based on the studies of suitability analysis, the results showed that a 0.9–0.95 fuzzy gamma coefficient has a high accuracy for the suitability map [53–55].

In the case of our study, overlaying the maps, we used a fuzzy γ -operator with a 0.9 gamma coefficient, and it is formulated as follows [52]:

$$\mu_{combination} = (\prod_{i=1}^{n} \mu_i)^{1-\gamma} (1 - \prod_{i=1}^{n} (1 - \mu_i))^{\gamma} \tag{4}$$

This "γ -operator" is a combination of the fuzzy algebraic product and the fuzzy algebraic sum, where γ is a parameter having values ranging from 0 to 1. If γ is 1, the combination is the same as the fuzzy algebraic sum while for a score of 0, the combination is equal to the fuzzy algebraic product. The parameter indicates where the actual operator is located between the logical ""and"" and ""or"" where μi is the fuzzy membership values for the i-th (i = 1, 2,. . ., n) maps to be integrated. The fuzzy algebraic sum operator is complementary to the fuzzy algebraic product. The result of this operation is always larger than, or equal to, the largest contributing fuzzy membership value. It means an increasing of possibility to obtain more suitable area.

**2.3.4. Defuzzification.**   The next step involved in the methodology is the defuzzification, which has two stages, a reclassification of the output of the model and generalization of final output into a final suitability map [56]. In the first step, we defuzzified it into six classes using the geographic equivalent classification method. Systematically evaluating the suitability of residential, the classification has been possibly explained as a linkage between urban development process and residential suitability. Moreover, the specification of the study area, the characteristic of the residential land was divided into two main features as apartment residential area (middle and high-rise residential buildings) and ger area (low-rise area with a house and traditional tent named as "ger" enclosed with fences) [3,57]. Finally, the classification of the degree of suitability was categorized as the most suitable, the second most suitable, moderately suitable, potentially suitable, marginally suitable and not suitable.

**2.3.5. Accuracy assessment of criteria map.** Several accuracy metrics can be obtained from the confusion matrix. The percentage of incorrectly classified (omitted) data and overall accuracy are the two metrics that are most widely recognized [58,59]. By dividing the total number of pixels by the number of correctly classified pixels in relation to the data used as ground truth, the overall accuracy score provides an evaluation. It's critical to understand the appropriate interpretation of each particular class in addition to the overall accuracy. The 'two standpoints' approach of Story and Congalton [60] is typically used to determine the correctness of each particular class: the producer's and user's accuracy. The number of cases appropriately allocated classes is divided by the total number of that case classes [59,60].

The K statistics is a commonly used metric to assess map accuracy that may be obtained from a confusion matrix. It provides an indication of the chance of agreement between the map and the reference data [61–66]. We are used as the reference data the randomly selected and visualized ground control points from the high-resolution satellite image of Google Earth (https://earth.google.com/web/) with field visited onsite manual evaluations.

Since the calibration of model not performed in the suitability evaluation steps, additionally employed the relative operating characteristic (ROC) curve and area under the ROC curve (AUC) value for the validation of criteria map [67,68]. AUC value (range of 0.5–1) close to 1 indicates perfect model with low bias and near 0.5 value shows low accuracy with random chance to predict. In practice, AUC values greater than 0.7 indicates that model/mapping has acceptable accuracy [69,70].

In our case, newly constructed or under construction residential blocks in the study area were counted as test data to evaluate accuracy of the produced suitability maps. All the processes of overall accuracy, K statistics and ROC curve (AUC) have been computed with the ArcGIS software (version 10.6) using ArcSDM tool (from Esri (http://www.arcgis.com).

## 3. Results

Overall, the spatial structure of the city is a monocentric while the spatial pattern of the residential suitability is polycentric. One main suitable area and four small area were found as the most suitable for living and formed a concentric ring pattern on the resulted map (Fig 3). Moreover, the result delineated that two main features of the most suitable residential area in the city, such as a high-rise residential area in the southern part of the city and a low-rise ger area in four small hotspots.

This study applied three main categories of suitability factors, including, livability, affordability, and accessibility from the assessment of 15 factors. Based on the final map's spatial pattern, criteria related to accessibility heavily influence the suitability, including, distance from bus stops, schools, kindergartens, hospitals, shopping centers, CBD and facility cover. Criteria of livability and affordability have relatively lower influences on the resulted suitability map.

The final map indicated the residential suitability (Fig 3). Approximately 3% (179.3 km$^2$) of the study area has suitability score above zero while the rest was identified as not suitable (Table 4).

However, the spatial pattern of Ulaanbaatar city is monocentric, a few small sub-centres have been created in the city (Fig 3). Moreover, the city centre (CBD) is highlighted as a not suitable for residential apartment purpose, instead, a few areas in the south were found to have potential. The residential suitability is decreasing from those hotspots with the distance. In terms of total area, 4.9 km$^2$ is identified as the most suitable for residential purpose consisting of 0.1% of the total identified suitable area. Approximately 80% of the area is suitable for a residential apartment on the most suitable (Table 4).

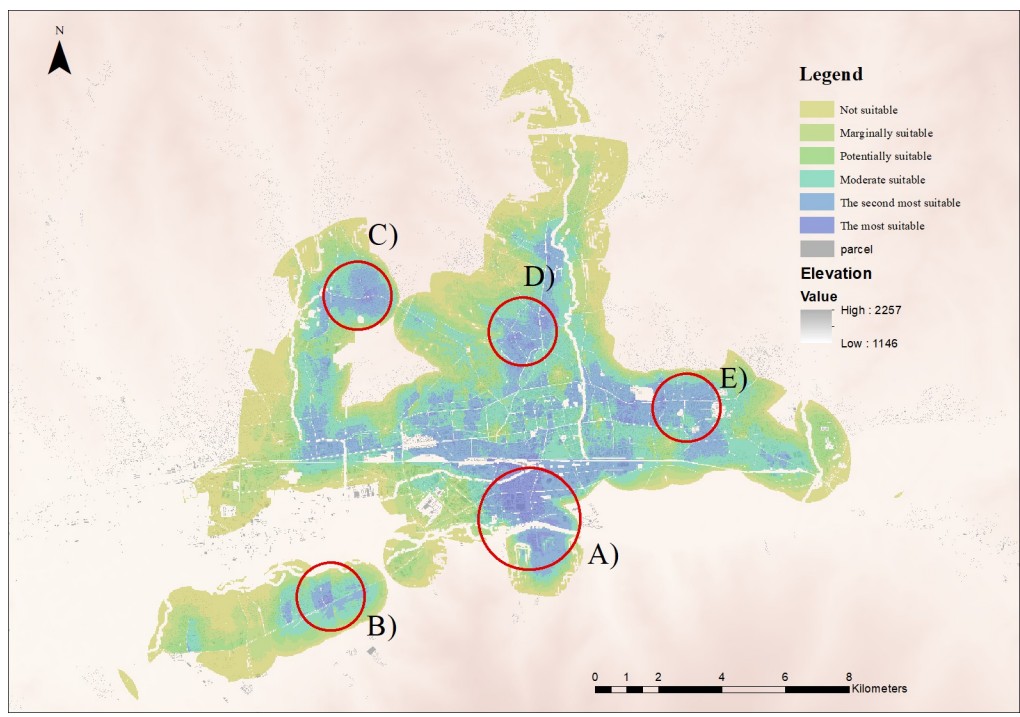

**Fig 3. Residential suitability map of Ulaanbaatar city.** Noted: (A) Most suitable apartment area, (B) Most suitable ger and apartment area, (C), (D), and (E) Most suitable ger area. The residential suitability map was created by the authors in QGIS (https://qgis.org/en/site/). Data source: The base map in this figure is from Geoportal of Mongolia's (ALMGC) parcel map [40] (Available online: https://nsdi.gov.mn/open-layer/), and DEM (Elevation) subset based on the radar-based SRTM elevation data from the CGIAR Consortium [43] (available online https://srtm.csi.cgiar.org).

The most suitable residential site in the apartment area has been found in the south and central part of the city. The boundary of this area is featured by the central railroad in the north, along the north-south axis road it's narrowing to south merging to the Tuul river creating the south-western edge, and central to south main avenue becomes the eastern boundary. It is administered as southwest district dominantly. The spatial pattern of the suitable area is clearly surrounding the most suitable area (Fig 3).

In terms of potential areas, they are more likely to be connected with the most suitable area of ger area. However, new suburb development in southwest district, it's the western, southern part of the area, is indicated as the most potential area.

Regarding the low-rise ger area, the number of the small hotspots is revealed on the map, and it is spatially clustered, unlikely to apartment suitability. In the north, locally known name

**Table 4. The area size resulted from analysis.**

| Classification of suitability | Area size (km²) | Cover rate |
| --- | --- | --- |
| Most suitable | 4.90 | 0.10% |
| Second most suitable | 32.29 | 0.68% |
| Moderate suitable | 42.29 | 0.89% |
| Potentially suitable | 29.78 | 0.63% |
| Marginally suitable | 29.50 | 0.62% |
| Not suitable | 40.65 | 0.86% |
| Not suitable | 4561.03 | 96.22% |

as traditional old black-market area and north transport hub of ger area are found as the most suitable as the ger area residential. Second, the central part of the north western, locally known as west-arm area, is identified as the most suitable in ger area. Finally, in the south of the west, locally known as new city centre is outputted as a high level of suitability.

With regards to the second most suitability, spatially, it particularly surrounded by the most suitable location of residential apartment with a monotonic decreasing function. Moreover, it is also found with a large extent of area from the east part, north, and western part. Similarly, regarding the ger area, the most suitable hotspots are surrounded by the suitable area. It is noted that the classification of "the suitable" area is well distributed and surrounding the CBD, and 32.3 km$^2$ area is calculated (0.7% of the study area).

An error matrix was calculated to provide more details for the accuracy assessment. The overall accuracy is 0.85, meaning that 85% of the pixels were identically classified in the model and in the reference data, the result of the residential suitability and the 61 ground control points (GCPs) from the high-resolution satellite image of Google Earth (https://earth.google.com/web/) visualization with field manual checks. This indicates that there is quite some good coincidence between the maps and GCPs. The Kappa coefficient also presents excellent agreement. A value of 0.82 indicates that the produced suitability map classes are about 82% better than a random assignment of these classes. However, it should be noted that the GCPs, besides random selection concerns, is also based on manual evaluation of the expert knowledge of authors, introducing same arbitrary criteria of performed model.

In the study, fuzzy AHP based residential suitability was divided into 6 classes. A total of 65 newly constructed or under construction residential blocks in the study area were selected as sample points (true positive) to evaluate accuracy of the produced suitability maps (false positive). ROC curve (AUC) has been computed with the ArcGIS software (version 10.6) using ArcSDM tool from Esri (http://www.arcgis.com) which is allowed to present graphically ROC curve and compute AUC value. Acceptable AUC value 0.74 indicates that performed suitability map has a sufficient degree of accuracy.

A reason for AUV value's near to 0.7 could be the differences on the newly erected residential block buildings not always found perfect suitable sites for construction. While, fulfilling market demand construction company, land developer face lack of land supply for production and have to follow urban development plans which not based on suitability. Planning experts based their analysis on existing physical constraint, feasibility and construction factor data to locate residential development zone which is far from suitability and customers' request.

## 4. Discussion

The study analyzed residential suitability in Ulaanbaatar city using fuzzy logic theory. As highlighted in the literature, the fuzzy logic theory was used to deal with suitability analysis in decision making [29,32]. It also involved more creativity than the traditional AHP approach [22]. The fuzzy AHP approach minimizes the subjective importance of the criteria and is the sophisticated method to deal with the uncertainty of an expert's knowledge bias. Ustaoglu and Aydinoglu [22] emphasized that urban suitability maps obtained from the fuzzy method provide a better interpretation of the land use characteristics. However, a more specific study is suggested for residential suitability analysis. Our finding suggested that the result from the selected method showing a competitive output.

Regarding the delineated boundary residential suitability area, approximately 3% of the city is identified with a suitability score from the lowest to highest (0–73), and it has created the residential suitability boundary (Fig 3). Apparently, this boundary closely matches with the boundaries that were found in other studies. JICA [7] highlighted the north edge of

prohibition urban development, and further from this boundary, it was identified to plan green area where is essential to the environment for sustainable city development. Erdenechimeg et al. [57] noted that the ger residential area categories as central area, mid-tier area, fringe area, apartment area and summer camp green space settlements. The boundary of the fringe ger area and summer camp green space settlements were almost matched with our findings. Moreover, a recent case study of urban sprawl in Ulaanbaatar city showed the suitable area of settlements, and the result is relatively similar with both the boundary and the sub-centres [71]. However, it can be said that those matching boundaries could be representing a weak feature of centralized infrastructure development, including road network, heating supply network, water supply network, sewerage network and the railroad network, and accessibility of basic services. Historically, the 'city's spatial structure is traditional monocentric, which tend to more equipped in the city centre in the Soviet era [9,10,49]. After the democratic revolution in 1990, a lack of urban development policies and immature land market during the unprecedented period with significant population growth pressured to form the uncontrolled and centralized formation of spatial structure in the city. However, it is unclear to define the border of the developing area of the city, caused of imprecise components and a subjective methodology. As a result, it enables a subjective influence on urban plans. Moreover, urban planners and decision-makers have acknowledged a circumstance that city problems related to environment, society, economy, transportation, infrastructure, and spatial planning are not possible to fully determine before the planning process. One of the prime steps in formulating the Ulaanbaatar master plan is "an evaluation of complex assessment for urban development". However, in this assessment, 65 factors of 4 categories, including environmental (26 factors), societal and economic (20 factors), infrastructural (8 factors), architectural, and landscape (8 factors) are evaluated, and the scoring is limited from 1 to 5 with arithmetic average [12]. Furthermore, the scoring of the evaluation did not recognize the spatial parameters (geometry). As a result, assessment is not possible due to inadequate information for spatial planning. Therefore, the proposed methodology should be made essential in planning projects, since our study's results clearly show the benefits of taking residential suitability analysis using fuzzy logic into consideration when examining suitability for new residential area development.

Furthermore, Bertaud [10] analyzed the spatial structure of Central and Eastern European (CEE) cities and found that it has a monocentric spatial structure causing a centralized command economy, maintaining their prestigious cultural centre, and converting to a polycentric structure slowly after their market transitions. Generally, one of the key changes in our results is that the monocentric spatial pattern is converting slowly into a polycentric structure (Fig 3). Regarding a high-rise apartment residential area, surprisingly, CBD is not identified as the most and second suitable site according to our findings, and the pattern of CBD has been changing to be mixed and converting to multi-purposes activities [3,6]. Moreover, the value of the residential area in CBD is slowly dropping due to inappropriate living condition, including a high density of building, underdeveloped public transportation, lack of green space and air pollution, indicating to develop and maintain in CBD as a cultural or business centre. UN [1] also noted this urbanization consequences, and government policy, regulation on urban development and investment responses market liabilities are the primary victims based on the CEE cities [1,9]. Moreover, the real estate market plays a vital role in city development, open information about prices, the business environment of real estate agent, clear land and property tenure, property registration institution, the flexibility of land use change, subsidies and taxation policy [49]. Ulaanbaatar city has demonstrated a similar spatial pattern of residential suitability. Since the economic capital of the citizens has increased, mid and high-societal groups are starting to prefer urban fringes or preurban agricultural areas for residential propose, compared to the socialist era, where CBDs were in high demand. Ulaanbaatar City's housing

market immediately reflected that kind of "close to nature" choice and private land developers started small residential development projects in suitable urban fringes, which are gradually changing an urban monocentric structure to a polycentric one.

It's evident, that Ulaanbaatar is the oldest urban center in Mongolia with a history of more than 700 years. The monocentric structure of the city had been created and administrated without land market implications during the Soviet era, with most urban growth of that time. This period saw land nationalization and centralized administrative allocation, impacting the internal spatial pattern of the city defined as prominent centers with extensive, radial, and concentric transit networks reinforcing their monocentric structure [72]. The combination of the centralized location of the national institutions, the main role in the domestic economy, and the high rate of urbanization in the post-socialist city founded insecure property rights, institutional disorganization, governance problems, and weak urban planning regulations. Mongolia's transition to a market economy was formed in the 1990s through the legal reform of numerous legal frameworks about land and property, including the Land Law, the Law on Allocation of Land to Citizens of Mongolia for Ownership, and the Law of Urban Development. Initially, the government proposed free land policy accelerated the market economy, helped the reduction of poverty, and allowed poor urban residents of slum districts (ger area) to own valuable residential land plots as immovable property. However, the free land policy harms urban management, accelerates urban expansion in city fringes, and creates difficulties in infrastructure construction, private owners refuse new development plans, increasing land speculation and corruption in the public sector.

The legal frameworks empower the local government of the capital city to organize land allocation, and public participatory planning principles are to be applied and reflected in urban development plans and activities [72–74]. Even though Mongolia has changed its urban regulation and administrative system, the historically centralized administration on the monocentric structure somehow exists and impacts the urban expansion on spatial patterns [9].

Regarding the most suitable area of apartment residential, the south and central part of the city is determined where indicates appropriate infrastructure, an accessible distance of basic services, CBD, and main park, a reasonable level of air pollution and crime rate (Fig 3). Moreover, the income level of residents in this suburb is sufficiently high with the poverty rate is less than 6%, according to the report of the World Bank [2].

With regards to ger area, four hotspots of the most suitable low rise ger area are the main indicators for the conversion of polycentric spatial structure. ADB [75] and JICA [7] recommended sub-centre locations on Ulaanbaatar Master Plan 2040, and they were almost the same locations as the most suitable hotspots of our study [12]. However, almost 40% of resident in ger area was identified as multidimensional poverty and high level of unemployment on The World Bank report [72]. Even though the location of those sites is the core of subregions as the most suitable ger residential, living condition is incomparably lower than apartment residential area (Fig 3). Furthermore, Singh [2] highlighted three dynamic factors for the low-density spatial pattern of Ulaanbaatar, including the traditional planning system and free land ownership, the unmatched urban plan with population growth after 2000. The main problem is the lack of centralized infrastructures, such as sewage, water supply, sanitary facilities, roads, and a public transportation system [57], and it is essential to improving living standard and filling the lack of basic services [75].

Regarding the potential for residential, the hotspots of most suitable for the low-rise ger area would be ideal for future developments. Also, for the high-rise residential area, the second most suitable category is preferable. Moreover, the number of locations would be the potential of suitability, if the condition of livability and accessibility are improved, including in low-rise ger area close to mid-rise residential area of CBD. Similarly, as the Ulaanbaatar city master

plan 2030, experts of JICA found and recommended that the restriction in the east, north and south for the urbanization of Ulaanbaatar city has to be taken in 2009 [7]. They only suggested promoting the urban expansion into the west due to environmental concerns in sustainable city development. Moreover, Gantumur et al. observed that the prediction of urban growth in 2030 and 2040 is projected toward the western and eastern part limited by the mountains [76]. Overall, based on the literature and our results, the potentiality of residential development depends on accessibility, and the southern and the western part of the city would be the most suitable for future development while some conditional development suggested the east and northern areas.

In term of the unsuitable residential area (99.93 km$^2$), it's expanding significantly [6,71] due to rural migration and free land policy implementation after 2002 [2,52]. The Law on Land (2002) grants Mongolian citizens, regardless of age and sex, one free land plot until May of 2028, in Ulaanbaatar city. According to the Land Inventory report [72], 277537 citizens owned land, and it was only 16.7% of the residents of the capital city. Despite the abundance of land plots owned by residents, there is still a significant need for 9715.5 hectares of land in Ulaanbaatar city [74]. The impact of free land policy is severe in the city, even though it is a market economy, residential ger area land is administratively allocated free for residents without a proper urban plan and basic infrastructures [72–74]. Once the land was allocated, it was almost never recycled since the difficulties of legislation for land-use recycle and urban development on the government decision [49]. The free land policy resulted in the absence of land prices, removing all economic incentives to the land use recycling process in the city [49,72].

As a result of this phenomenon, low-rise ger area continuously expands in the urban periphery with a low living condition, accelerating multidimensional poverty. Moreover, it is forming a spatial pattern of a concentric ring in the city and added by this process. Similarly, in Moscow and CEE cities, this phenomenon was observed and noted in the persistence and uniformity of housing types in successive rings [9,49]. Furthermore, whether in ger area or in sub-centre, households tend to be concentrated in the periphery with a low developed infrastructure, this spatial pattern is led to the increase of costs of living and pollutions.

## 5. Conclusions

To conclude, key changes are observed from the results, and it can be said that the city spatial pattern of residential suitability is changing to polycentric structure slowly. Firstly, the spatial pattern of residential suitability in Ulaanbaatar city is in a stage of an ongoing process of sub-urbanization, which indicates that the centre of the city (CBD) is not suitable for residential purpose. Secondly, the most suitable is only an area of 4.9 km$^2$, and almost 80% of the category was determined as a high-rise residential area, found in the southern part of the city. Thirdly, approximately 3% (179.3 km$^2$) of the study area is evaluated as a suitable while the rest is identified as unsuitable. Moreover, the ger area is the most important part of the spatial pattern of residential suitability, the specific living condition to consider for further urban development. Moreover, the ger area tends to expand through the progressive addition of concentric rings in the periphery of the city due to the distortion of the land market.

Further study should use more accurate data and identify the necessity of basic services (school, kindergarten, bus stop, hospital, public services, social services, green space, playground), road, central infrastructure access (water, heating, transport, sanitary) on each suburb.

## Supporting information

**S1 Fig. AUC of suitability map.** AUC is computed with the ArcSDM tool of ArcGIS software (version 10.6) from ESRI (http://www.arcgis.com).
(TIF)

**S1 Table. Accuracy assessment result.**
(XLSX)

## Acknowledgments

The authors acknowledge all the data providers, especially the General Agency for Land Management, Geodesy and Cartography of Mongolia, Research Laboratory of Land Planning and Survey, National University of Mongolia and JICA. Lastly, we would like to express our sincere gratitude to Jianhong (Cecilia) Xia, the editors and two anonymous reviewers for their constructive suggestions, which greatly raised the quality of the paper.

## Author Contributions

**Conceptualization:** Galmandakh Boldbaatar, Gantulga Gombodorj, Myagmartseren Purevtseren.

**Data curation:** Galmandakh Boldbaatar, Gantulga Gombodorj, Dorligjav Donorov, Robert Andriambololonaharisoamalala, Myagmarjav Indra.

**Formal analysis:** Galmandakh Boldbaatar, Gantulga Gombodorj, Robert Andriambololonaharisoamalala.

**Funding acquisition:** Myagmartseren Purevtseren.

**Investigation:** Dorligjav Donorov, Myagmarjav Indra.

**Methodology:** Galmandakh Boldbaatar, Gantulga Gombodorj, Robert Andriambololonaharisoamalala, Myagmarjav Indra.

**Project administration:** Myagmartseren Purevtseren.

**Resources:** Gantulga Gombodorj.

**Software:** Dorligjav Donorov, Robert Andriambololonaharisoamalala, Myagmarjav Indra.

**Supervision:** Myagmartseren Purevtseren.

**Validation:** Myagmarjav Indra, Myagmartseren Purevtseren.

**Visualization:** Dorligjav Donorov.

**Writing – original draft:** Galmandakh Boldbaatar, Gantulga Gombodorj, Robert Andriambololonaharisoamalala.

**Writing – review & editing:** Myagmarjav Indra, Myagmartseren Purevtseren.

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
