## [Decision Letter · Decision Letter 0]

10 May 2024

PONE-D-24-12830Land suitability analysis in monocentric post-socialist city: case of Ulaanbaatar, MongoliaPLOS ONE

Dear Dr. Purevtseren,

Thank you for submitting your manuscript to PLOS ONE. After careful consideration, we feel that it has merit but does not fully meet PLOS ONE’s publication criteria as it currently stands. Therefore, we invite you to submit a revised version of the manuscript that addresses the points raised during the review process.

We look forward to receiving your revised manuscript.

Kind regards,

Dr. Walid Al-Shaar

Academic Editor

PLOS ONE

Journal Requirements:

"This research was funded by the National University of Mongolia (NUM), research project and grant number: P2022-4385 and P2023-4621"

3. We note that Figures 1, 2 and 3 in your submission contain [map/satellite] images which may be copyrighted. All PLOS content is published under the Creative Commons Attribution License (CC BY 4.0), which means that the manuscript, images, and Supporting Information files will be freely available online, and any third party is permitted to access, download, copy, distribute, and use these materials in any way, even commercially, with proper attribution. For these reasons, we cannot publish previously copyrighted maps or satellite images created using proprietary data, such as Google software (Google Maps, Street View, and Earth). For more information, see our copyright guidelines: http://journals.plos.org/plosone/s/licenses-and-copyright.

a. You may seek permission from the original copyright holder of Figures  1, 2 and 3 to publish the content specifically under the CC BY 4.0 license.  

Reviewers' comments:

Reviewer's Responses to Questions

**Comments to the Author**

1. Is the manuscript technically sound, and do the data support the conclusions?

Reviewer #1: Yes

Reviewer #2: Yes

2. Has the statistical analysis been performed appropriately and rigorously? 

Reviewer #1: Yes

Reviewer #2: Yes

3. Have the authors made all data underlying the findings in their manuscript fully available?

Reviewer #1: Yes

Reviewer #2: Yes

4. Is the manuscript presented in an intelligible fashion and written in standard English?

Reviewer #1: Yes

Reviewer #2: Yes

5. Review Comments to the Author

Reviewer #1: - Identify 3-5 keywords under the paragraph of the abstract

- In the introduction part, the citations in years 2018-2020 are quite too old. Please find the more updated papers to support the paragraph of background of the study and problem statement.

- Please recheck the quality of all figures in the paer becase they are not appeared in the file submitted.

Reviewer #2: Generally, the paper provides an alternative analysis of land suitability, especially for former socialist cities.

The following suggestions/clarifications would enhance the impact of the research

1) Explain how land suitability analysis contributes to understanding or resolving the problems related to particular post-socialist cities.

2) Lines 158-163, the crime rate is classified into five classes based on the parameters used in Table 3. However, there is no explanation or explanation for the assumption you took in order to define a category as "high" or "very high" as in the case of crime rate "high" belongs to 0-0.6, why not 0-0.5? The selection of your parameters and how they are valued is an important aspect of your research. The sources for Table 3 are also lacking

3)      Line 212.. Could you please explain why the gamma coefficient is 0.9?

4)      Line 411, what is the free land policy? Can you describe how it significantly affects the urban expansion and relate it to your results?

5)      On line 426 of your conclusion, you mentioned that the city spatial pattern of residential suitability is changing, however you did not describe how it has changed over time.

6)      As this is a land suitability analysis study, one would expect to see more of a focus on how post- or pre-socialist urban policies contribute to land expansion and suitability. In this manner, policymakers will be able to use the findings to formulate strategies that can be implemented.

6. PLOS authors have the option to publish the peer review history of their article (what does this mean?). If published, this will include your full peer review and any attached files.

Reviewer #1: **Yes: **Asst.Prof.Dr.Wissawa Aunyawong

Reviewer #2: **Yes: **Yilak Kebede

---

## [Author Response · Author response to Decision Letter 0]

8 Jul 2024

Dear Reviewers,

Thank you very much for the opportunity to submit our revised manuscript titled “Land suitability analysis in monocentric post-socialist city: case of Ulaanbaatar, Mongolia” co-authored with Galmandakh Boldbaatar, Gantulga Gombodorj, Dorligjav Donorov, Robert Andriambololonaharisoamalala, Myagmarjav Indra, and Myagmartseren Purevtseren to PLOS ONE. 

We are extremely grateful to the two anonymous reviewers for providing detailed, thoughtful, and helpful comments, which led us to reconsider the explanations of the analysis, refocus the literature review and strengthen the paper. We believe that we were able to address almost all comments made by the reviewers. In our reply to reviewers, we explain how we addressed each comment.

We have addressed all the comments that are now reflected in our revised manuscript and inserted line numbers and highlighted significant changes by using GREEN text highlight color to expedite the checking process.

Here is a point-by-point response to the reviewers’ comments and concerns.

Response for Reviewer #1

We appreciate the reviewers’ comments, which we find to be constructive and delivered professionally and appropriately. Thank you very much for your comments, each of which is addressed in the following. Our revised manuscript reflects your comments and suggestions. We have addressed all the comments that are now reflected in our revised manuscript and inserted line numbers and highlighted significant changes by using GREEN text highlight color to expedite the checking process. 

Comments:

1- Identify 3-5 keywords under the paragraph of the abstract

Response: Thank you for your detailed and well-thought-out review. Our detailed responses to your criticisms and suggestions are provided below.

Action: Keywords added in Line 44-45 as below:

Keywords: urban multicriteria analysis; suitability evaluation; fuzzy logic; urban development; sustainable city

Comments:

2- In the introduction part, the citations in years 2018-2020 are quite too old. Please find the more updated papers to support the paragraph of background of the study and problem statement.

Response: We thank you for this valuable comment. Following your kind advice, the revised manuscript engages more strongly with existing findings published in interdisciplinary journals in recent years. Our introduction to the existing literatures now consists of more recent literatures.

Action: New citations added Line 106, 115, 139 as below:

Therefore, it is now widely admitted that analyzing and identifying the suitability of residential areas is very important for appropriate land use policies to improve the living condition of citizens for future urban development [9, 10, 15]. 

The process of land suitability analysis involved multicriteria analysis (MCA) and geographic information system (GIS), and the land suitability evaluation methods have significantly improved due to the development of data science and GIS. Multicriteria analysis manages complex factors for suitability analyses and has been used in various industries [16,17,18,19]. 

To minimize the subjectivity and mathematical shortcomings, the fuzzy logic theory, which was initially developed by Zadeh [34], was highly valued in the latest studies [24,35,36]. 

New citations added on Reference in line 600, 608-614, 660-667 as below:

References

15. Kazemi, F., & Hosseinpour, N. (2022). GIS-based land-use suitability analysis for urban agriculture development based on pollution distributions. Land Use Policy, 123, 106426. https://doi.org/10.1016/j.landusepol.2022.106426

18. Andrea Zaniboni, Patrizia Tassinari, Daniele Torreggiani, GIS-based land suitability analysis for the optimal location of integrated multi-trophic aquaponic systems, Science of The Total Environment, Volume 913, 2024, https://doi.org/10.1016/j.scitotenv.2023.169790.

19. Mohamed E.M. Jalhoum, Mostafa A. Abdellatif, Elsayed Said Mohamed, Dmitry E. Kucher, Mohamed Shokr, Multivariate analysis and GIS approaches for modeling and mapping soil quality and land suitability in arid zones, Heliyon, Volume 10, Issue 5, 2024, https://doi.org/10.1016/j.heliyon.2024.e27577.

35. Sarasie Tennakoon, Armando Apan, Tek Maraseni, Richard Dein D. Altarez, Decoding the impacts of space and time on honey bees: GIS based fuzzy AHP and fuzzy overlay to assess land suitability for apiary sites in Queensland, Australia, Applied Geography, Volume 155, 2023, https://doi.org/10.1016/j.apgeog.2023.102951.

36. Lian Xue, Peng Cao, Deze Xu, Ying Guo, Qingfang Wang, Xingfei Zheng, Ruijuan Han, Aiqing You, Agricultural land suitability analysis for an integrated rice–crayfish culture using a fuzzy AHP and GIS in central China, Ecological Indicators, Volume 148, 2023, https://doi.org/10.1016/j.ecolind.2022.109837.

Comments:

3- Please recheck the quality of all figures in the paer becase they are not appeared in the file submitted.

Response: We thank you for this suggestion. Please see inserted figures in main body text: figures in line 169 (Fig 1), in line 265 (Fig 2) and in line 392 (Fig 3)

Action: Three figures inserted in the article and as well attached on article submission system were upgraded. They have now more resolution and good quality.

Response for Reviewer #2

We are very grateful to reviewer 2 for detailed and thoughtful comments which we find to be constructive and delivered professionally and appropriately. Thank you very much for your comments, each of which is addressed in the following. We have addressed all the comments that are now reflected in our revised manuscript and inserted line numbers and highlighted significant changes by using GREEN text highlight color to expedite the checking process.

Comments:

1) Explain how land suitability analysis contributes to understanding or resolving the problems related to particular post-socialist cities.

Response: Thank you very much for the constructive comment. It has helped us to significantly improve the introduction section of the paper.

Action: Added and modified in Line 68-103 as below:

Furthermore, the typical monocentric spatial arrangement of the city was increasingly equipped in the city center throughout the Soviet era [9]. A significant population increase following the democratic revolution in 1990 created unregulated pressure on the city's centralized spatial structure. Additionally, Bertaud [10] examined the spatial organization of cities in Central and Eastern Europe (CEE) and discovered that these regions have a monocentric spatial structure. Following market transitions, it observed a gradual transition to polycentric structures. Despite limited research on the monocentric spatial structure of Ulaanbaatar city, there is a noticeable presence of this kind of transition in this area..

Urban planning is consistently linked to a variety of decision-making difficulties and strategies, and the results of planning will differ based on the combination of the approaches and techniques employed. During the Soviet era, Mongolia, like many other communist nations, employed planning approaches and techniques to design cities without involving citizens. This resulted in a simple, quick, and interruption-free planning procedure during the project [11].

In our knowledge, in the Soviet era, a monocentric structure allowed the communist government to “control everything” related to urban management. According to the centralized economy, it was cost-effective and easy to manage particularly to build mega infrastructure and new residential district. The urban development process was implemented under “bulldozer urbanism” approaches, whereas all land had been owned by the state. There was no need for residents' consent or negotiations with them to expropriate their lands. The municipality directly canceled the resident permit and gave notice to vacate the new development area by official declaration. This procedure referred to as “enter to ground” characterizes communist regimes' relation to their people. For the new development areas, only the land capability for construction was considered, focusing on investigating the physical characteristics of ground soil and topography. 

These days, many of these nations have new laws requiring public engagement while also using new approaches and technologies. One of the most difficult tasks facing communities and municipal governments today is to locate prospective locations for the sustainable growth of city subcenters without threatening human rights and environmental protection. Land suitability analysis offers tools to address these management problems [11]. 

Ulaanbaatar city’s monocentric structure hinders sustainable development promoting traffic jams, air pollution, land grabbing, etc. Several international and local experts suggested to replan and switch from monocentric to polycentric structures by developing new city subcenters in Ulaanbaatar [7,11,12]. 

In short, there has been a constant increase in interest in adopting land suitability analysis since it can handle and combine many types of data (both spatial and non-spatial, as well as multi-temporal and multi-scale) in an efficient manner. Then, from a technical perspective, comprehension of the fundamental issue, and democratic points of view can all benefit the planning process [11].

Comments:

2) Lines 158-163, the crime rate is classified into five classes based on the parameters used in Table 3. However, there is no explanation or explanation for the assumption you took in order to define a category as "high" or "very high" as in the case of crime rate "high" belongs to 0-0.6, why not 0-0.5? The selection of your parameters and how they are valued is an important aspect of your research. The sources for Table 3 are also lacking.

Response: Thank you very much for the thoughtful comment. We followed your comment and added explanation on crime rate classification with used sources. The sources of Table 3 are added under it. 

Action: Added and modified in Line 213-215 as below:

The crime maps data are categorized using a modified version of the Jenks natural breaks classification method, aiming to minimize within-group variance and maximize between-group variance [44]. Crime rate shows a percentage of recorded crime per population in each suburb and was categorized into five classes from low to high. 

In line 688 in the reference:

44. Chen, J., Yang, S. T., Li, H. W., Zhang, B., and Lv, J. R.: Research on Geographical Environment Unit Division Based on the Method of Natural Breaks (Jenks), Int. Arch. Photogramm. Remote Sens. Spatial Inf. Sci., XL-4/W3, 2013. 47–50, https://doi.org/10.5194/isprsarchives-XL-4-W3-47-2013,

In line 256-257: 

All the classification methods of criteria are based on literature sources in Table 2 and below presented criteria classifications in detail (Table 3).

In line 255:

Table 3. Parameters used in the analysis of residential suitability. 

Data sources: Crime rate were generated from the open data of NSO’s website [43], land use, facility cover, building cover, land price, distance from main park, distance from water protection zones, distance from shopping centre were provided by ALMGC through the open-spatial data online portal [40], distance from air pollution point, distance from school, kindergarten, hospital, bus stop are processed data of the NUM-Lab.RaS [42], slope degree information was gathered from a subset of an optical stereo-based ASTER GDEM elevation data as well as radar-based SRTM elevation data [39,41], distance from CBD information based on JICA study [7]. 

 Comments: 

3) Line 212. Could you please explain why the gamma coefficient is 0.9?

Response: Thank you for your detailed and well-thought-out review. We followed the reviewers’ comment and added explanation on gamma coefficient number with based literature sources.

Action: Added and modified in Line 278-280 as below:

Zimmermann and Zysno [47] discussed a variety of fuzzy overlay operations and highlighted five useful operators to combine independent datasets with the fuzzy AND, fuzzy OR, fuzzy algebraic product, fuzzy algebraic sum and fuzzy γ -operators. Based on the studies of suitability analysis, the results showed that a 0.9-0.95 fuzzy gamma coefficient has a high accuracy for the suitability map [53, 54, 55].

In Line 713-722 at the reference list

53. Kumar, R., Anbalagan, R. Landslide susceptibility zonation in part of Tehri reservoir region using frequency ratio, fuzzy logic and GIS. J Earth Syst Sci 124, 431–448 (2015). https://doi.org/10.1007/s12040-015-0536-2.

54. Baharvand, S., Rahnamarad, J., Soori, S. et al. Landslide susceptibility zoning in a catchment of Zagros Mountains using fuzzy logic and GIS. Environ Earth Sci 79, 204 (2020). https://doi.org/10.1007/s12665-020-08957-w.

55. Djouher Saadoud, Mohamed Hassani, Francisco José Martin Peinado, Mohamed Saïd Guettouche, Application of fuzzy logic approach for wind erosion hazard mapping in Laghouat region (Algeria) using remote sensing and GIS, Aeolian Research, Volume 32, 2018, Pages 24-34, https://doi.org/10.1016/j.aeolia.2018.01.002.

Comments:

4) Line 411, what is the free land policy? Can you describe how it significantly affects the urban expansion and relate it to your results?

Response: We thank you for this valuable comment and for pointing us to do the clear explanation on the result of our study. It has helped us to significantly improve the paper.

Action: Added and modified in Line 495-513 as below:

In term of the unsuitable residential area (99.93 km2), it’s expanding significantly [6,71] due to rural migration and free land policy implementation after 2002 [2,52]. The Law on Land (2002) grants Mongolian citizens, regardless of age and sex, one free land plot until May of 2028, in Ulaanbaatar city. According to the Land Inventory report [72], 277537 citizens owned land, and it was only 16.7 % of the residents of the capital city. Despite the abundance of land plots owned by residents, there is still a significant need for 9715.5 hectares of land in Ulaanbaatar city [74]. The impact of free land policy is severe in the city, even though it is a market economy, residential ger area land is administratively allocated free for residents without a proper urban plan and basic infrastructures [72,73,74]. Once the land was allocated, it was almost never recycled since the difficulties of legislation for land-use recycle and urban development on the government decision [49]. The free land policy resulted in the absence of land prices, removing all economic incentives to the land use recycling process in the city [49, 72]. 

As a result of this phenomenon, low-rise ger area continuously expands in the urban periphery with a low living condition, accelerating multidimensional poverty. Moreover, it is forming a spatial pattern of a concentric ring in the city and added by this process. Similarly, in Moscow and CEE cities, this phenomenon was observed and noted in the persistence and uniformity of housing types in successive rings [9,49]. Furthermore, whether in ger area or in sub-centre, households tend to be concentrated in the periphery with a low developed infrastructure, this spatial pattern is led to the increase of costs of living and pollutions.

New citations added in Line 766-775 as Reference and as below 

72. The World Bank, Land Administration and Management in Ulaanbaatar, Mongolia, final report, 2018, Washington, USA, Available online: https://www.worldbank.org/en/country/mongolia/publication/land-administration-and-management-in-ulaanbaatar-mongolia.

73. The General authority for Land administration, geodesy and cartography of Mongolia, Land Inventory Report of Mongolia, 2024, Ulaanbaatar, Mongolia, Available online: https://egazar.gov.mn/api/landuse/nav.

74. Gombodorj G, 2023, 20 years of Land ownership of Mongolia (power point presentation), the conference of 20 years anniversary of Land ownership, Ulaanbaatar city Government Palace, Mongolia.

Comments:

5) On line 426 of your conclusion, you mentioned that the city spatial pattern of residential suitability is changing, however you did not describe how it has changed over time.

Response: Thank you for bringing up the interesting idea to describe residential suitability change in details.

---

## [Decision Letter · Decision Letter 1]

30 Jul 2024

Land suitability analysis in monocentric post-socialist city: case of Ulaanbaatar, Mongolia

PONE-D-24-12830R1

Dear Dr. Purevtseren,

We’re pleased to inform you that your manuscript has been judged scientifically suitable for publication and will be formally accepted for publication once it meets all outstanding technical requirements.

Kind regards,

Dr. Walid Al-Shaar

Academic Editor

PLOS ONE

Additional Editor Comments (optional):

Reviewers' comments:

Reviewer's Responses to Questions

**Comments to the Author**

1. If the authors have adequately addressed your comments raised in a previous round of review and you feel that this manuscript is now acceptable for publication, you may indicate that here to bypass the “Comments to the Author” section, enter your conflict of interest statement in the “Confidential to Editor” section, and submit your "Accept" recommendation.

Reviewer #1: All comments have been addressed

Reviewer #2: All comments have been addressed

2. Is the manuscript technically sound, and do the data support the conclusions?

Reviewer #1: Yes

Reviewer #2: Yes

3. Has the statistical analysis been performed appropriately and rigorously? 

Reviewer #1: Yes

Reviewer #2: Yes

4. Have the authors made all data underlying the findings in their manuscript fully available?

Reviewer #1: Yes

Reviewer #2: Yes

5. Is the manuscript presented in an intelligible fashion and written in standard English?

Reviewer #1: Yes

Reviewer #2: Yes

6. Review Comments to the Author

Reviewer #1: I appreciate for the revised manuscript. Now, it engages more strongly with existing findings published in interdisciplinary journals in recent years. The introduction to the existing literatures now consists of more recent literatures.

Reviewer #2: (No Response)

7. PLOS authors have the option to publish the peer review history of their article (what does this mean?). If published, this will include your full peer review and any attached files.

Reviewer #1: **Yes: **Asst.Prof.Dr.Wissawa Aunyawong

Reviewer #2: **Yes: **Yilak Kebede

---

## [Editor Report · Acceptance letter]

6 Aug 2024

PONE-D-24-12830R1 

PLOS ONE

Dear Dr. Purevtseren, 

I'm pleased to inform you that your manuscript has been deemed suitable for publication in PLOS ONE. Congratulations! Your manuscript is now being handed over to our production team.

Kind regards, 

on behalf of

Dr. Walid Al-Shaar 

Academic Editor

PLOS ONE